

# Identification of novel prophage regions in *Xenorhabdus nematophila* genome and gene expression analysis during phage-like particle induction

Emilie Lefoulon[1], Natalie Campbell[1] and S. Patricia Stock[1,2]

[1] School of Animal and Comparative Biomedical Sciences, University of Arizona, Tucson, AZ, USA

[2] College of Agriculture, California State University, Chico, CA, USA

Corresponding author
S. Patricia Stock,
spstock@csuchico.edu

## ABSTRACT

**Background:** Entomopathogenic *Xenorhabdus* bacteria are endosymbionts of *Steinernema* nematodes and together they form an insecticidal mutualistic association that infects a wide range of insect species. *Xenorhabdus* produce an arsenal of toxins and secondary metabolites that kill the insect host. In addition, they can induce the production of diverse phage particles. A few studies have focused on one integrated phage responsible for producing a phage tail-like bacteriocin, associated with an antimicrobial activity against other *Xenorhabdus* species. However, very little is known about the diversity of prophage regions in *Xenorhabdus* species.

**Methods:** In the present study, we identified several prophage regions in the genome of *Xenorhabdus nematophila* AN6/1. We performed a preliminary study on the relative expression of genes in these prophage regions. We also investigated some genes (not contained in prophage region) known to be involved in SOS bacterial response (*recA* and *lexA*) associated with mitomycin C and UV exposure.

**Results:** We described two integrated prophage regions (designated Xnp3 and Xnp4) not previously described in the genome of *Xenorhabdus nematophila* AN6/1. The Xnp3 prophage region appears very similar to complete Mu-like bacteriophage. These prophages regions are not unique to *X. nematophila* species, although they appear less conserved among *Xenorhabdus* species when compared to the previously described p1 prophage region. Our results showed that mitomycin C exposure induced an up-regulation of *recA* and *lexA* suggesting activation of SOS response. In addition, mitomycin C and UV exposure seems to lead to up-regulation of genes in three of the four integrated prophages regions.

## INTRODUCTION

Bacteria employ diverse strategies to occupy different ecological niches and outcompete distantly related or kin bacteria (*Griffin, West & Buckling, 2004*). Among them, is bacteriophage DNA, (prophages or their remnants) which is integrated in bacterial

genomes (*Vacheron, Heiman & Keel, 2021*). These prophage regions present costs and benefits. Potential costs can be the re-entry into the lytic cycle of the phage causing the bacterial death, or energy costs associate with the maintenance of additional genetic material prophages for the bacteria (*Harrison & Brockhurst, 2017*; *Selva et al., 2009*). However, prophages can be beneficial by adding diversity to the bacterial gene repertoires or act as weapons for bacterial competition (*Harrison & Brockhurst, 2017*). Indeed, prophages can be co-opted by bacteria for production of derivatives or phages, such as phage-like bacteriocins (*Bobay, Touchon & Rocha, 2014*; *Vacheron, Heiman & Keel, 2021*).

*Xenorhabdus* spp. are intriguing bacteria with a dual lifecycle, they are endosymbionts of entomopathogenic *Steinernema* nematodes and together they form a pathogenic partnership that kills a wide range of insect species. The infective juvenile stages of the nematodes harbor bacteria in an intestinal receptacle protecting them from the external environment (*Flores-Lara et al., 2007*; *Kim, Flores-Lara & Stock, 2012*), and vector them from one insect host to another (*Boemare, 2002*). Once inside an insect host, *Steinernema* nematodes release *Xenorhabdus* symbionts, which produce antimicrobial secondary metabolites that preserve the cadaver from decomposing allowing the nematodes to reproduce and complete their life cycle (*Stock, 2019*; *Stock & Goodrich-Blair, 2008*).

The arsenal of toxins produced by *Xenorhabdus* is not limited to secondary metabolites. The production of phage-like particles were reported for the first time by *Poinar, Hess & Thomas (1980)* and described as "defective phages" composed of an inner core and a contractile sheath caring a base plate. Studies showed that these tail-like phage particles (named xenorhabdicin) exhibit bactericidal activity against closely related bacterial species (*Poinar, Hess & Thomas, 1980*; *Thaler, Baghdiguian & Boemare, 1995*). It has been suggested that these xenorhabdicins may constitute another antimicrobial barrier to maintain the symbiotic association (*Thaler, Baghdiguian & Boemare, 1995*) or might be involved in interspecies competition (*Morales-Soto & Forst, 2011*). Although, diverse phage particles were also described in *Xenorhabdus nematophila* (*Boemare et al., 1992*; *Thaler, Baghdiguian & Boemare, 1995*), most of the research has focused on xenorhabdicins (*Ciezki et al., 2017*; *Morales-Soto & Forst, 2011*; *Thappeta et al., 2020*). Studies showed that a particular prophage region integrated in *Xenorhabdus* spp. genomes is involved in the induction of these xenorhabdicin (*Morales-Soto et al., 2012*; *Morales-Soto & Forst, 2011*). Two distinct prophages regions have been described in the genomes of *X. nematophila* and *Xenorhabdus bovienii*: one of these regions "p1 prophage" (respectively *xnp1* or *xbp1*) contains tail synthesis genes but lacks genes for capsid synthesis and replication, and the second region named p2 (*xnp2 or xbp2*) is similar to an intact P2 prophage (*Morales-Soto et al., 2012*; *Morales-Soto & Forst, 2011*). Only the *xnp1* prophage region appears to be up-regulated by exposure to mitomycin C (*Morales-Soto & Forst, 2011*). In addition, in *X. bovienii* and *X. nematophila* inactivation of the sheath gene of the *xnp1* or *xbp1* region results in the loss of phage tail-like bacteriocin activity (*Morales-Soto & Forst, 2011*). Although it has been suggested that *X. bovienii* carries several other prophage clusters (more similar to *lambda* and *Mu* prophage) (*Ciezki et al., 2017*) no further investigations have been performed to confirm this statement.

Herein, we focus on the discovery of other prophage regions in of *X. nematophila* AN6/1 genome and conducted a preliminary study on the expression of in different prophage regions during mitomycin C and UV exposure. We choose to compare the effect of the mitomycin C the study of *Xenorhabdus* prophage regions (*Boemare et al., 1992*; *Ciezki et al., 2017*; *Morales-Soto & Forst, 2011*; *Thaler, Baghdiguian & Boemare, 1995*; *Thappeta et al., 2020*) and, the effect of UV which has been previously used to induced phage particles on other bacteria (such as *Escherichia coli*) (*Baluch & Sussman, 1978*; *Takebe et al., 1967*).

## MATERIALS AND METHODS

### Prophage region analysis

Prophage regions present in the genomes of *Xenorhabdus* spp. were identified using PHASTER (PHAge Search Tool Enhanced Release) (*Arndt et al., 2016*). PHASTER classified and detected prophage region by completeness (incomplete, questionable and intact) thus we focused on regions with higher score detected as "intact" (*Arndt et al., 2016*). Annotation of the proteins localized in these regions were confirm by homology searches between the protein sequences using blastp (*Camacho et al., 2009*) and hidden Markov Models (HMM) profile using hmmscan in the HMMER web server (*Potter et al., 2018*).

### Bacterial strain, culture conditions and mitomycin C-induced or UV-induced cultures

Cultures of *Xenorhabdus nematophila* strain AN6/1 were considered in the current study that are maintained in P. Stock's laboratory (University of Arizona). The bacteria were grown in 10 mL Luria Bertani (LB) broth supplemented with 0.1% (w/v) sodium pyruvate (LBP) in a 50 ml centrifuge tube with agitation at 28 °C overnight for 12–16 h. Culture growth was monitored based on 600 nm optical density (OD600). Then, the bacteria were subculture (1:5 dilution) and grown at 28 °C to reach OD600 of 0.4–0.5. Two types of induction were tested: (a) with mitomycin C at a concentration of 5 µg ml$^{-1}$ and (b) 10 min exposure to UV 6 µJ/cm$^2$. Negative controls consisted of cultures not induced. Both induced and uninduced cultures were incubated at 28 °C in an incubator with moderate shaking (1 g). Two independent experiments were performed to assess: (1) late effect of induction with two studied time points: 2 and 10 h after induction, and (2) early effect of mitomycin C induction with two studied time points: 15 and 40 min after induction. For both experiments, OD600 was quantified at each time point and results were plotted using scatterplot in R environment (*Fox & Weisberg, 2019*).

### RNA extraction and RT-PCR

At each time point, three replicates of 1.5 mL of cultures were collected, centrifuged at 3,080 g for 10 min. The supernatant was discarded and 750 µL Trizol was added to the pellet. Then, 150 µL chloroform was added to the bacteria mix and incubated on ice 15 min. After centrifugation at (20,821 g for 15 min at 4 °C) the aqueous phase was

transferred to a new tube and 375 μL of isopropanol were added. After 10 min of incubation on ice, samples were centrifuge for 10 min at 20,821 g at 4 °C. The supernatant was discarded, and 1 mL of 75% ethanol was added to rinse the samples which were then vortexed, followed by centrifugation for 5 min at 7,885 g at 4 °C. The rinse step was repeated, and the RNA was air-dry and resuspend in water RNase-free. An additional incubation step for 10 min at 55 °C was considered to improve yield of the RNA elution. The RNA was quantified using a Nanodrop and 300 ng of RNA was used as the template for cDNA synthesis. cDNA was synthesized using the Bioline SensiFast cDNA Synthesis kit following manufacturer's protocols.

## qPCR of prophage region genes

Quantitative real-time polymerase chain reaction (qRT-PCR) was used to measure the relative gene expression across induction conditions. Each primer set condition was optimized using Bioline SensiFast No ROX Sybr Master Mix and summarized in the Table S1. Primers were designed based on the genome reference of *Xenorhabdus nematophila* AN6/1 (NZ_LN681227) using AmplifX (v2.0.7) (Nicolas Jullien; https://inp.univ-amu.fr/en/). A total of 10 pairs primers were used in this study (Table S1). We normalized the gene expression using the DNA gyrase subunit A, *gyrA* as housekeeping gene. We used the Pfaffl model to calculate the relative gene expression using the formula: $\frac{(E_{GOI})^{CtGOI}}{(E_{Ref})^{CtRef}}$ (*Pfaffl, 2001*). Primer efficiency was determined for each gene by running a standard curve and converted primers efficiency (E) was calculated as follows: (primer efficiency (%)/100) + 1. The calculated Ct from uninduced cultures was used as a calibrator when calculating the ΔCt values for all the samples. The relative gene expression values were transformed by a logarithmic base 2 function to be plot as Boxplot in R environment. The distribution of the data was not normal, thus the variance of the relative expression with the non-parametric Kruskal-wallis rank sum associated with multiple pairwise comparisons was considered using the Dunn's test in the R environment (dunn.test package) (*Dunn, 1964*).

## Phage particles preparation and SDS-PAGE gels

Three time points were selected to purified induced phage particles: the two time points of the early induction experiment, 15 and 40 min after induction, and the last time point of the late induction experiment (10 h after induction). A total of 8 mL of induced and uninduced cultures were centrifuged at 3,080 g for 10 min to remove cellular debris. The supernatants were filtered using 0.22 μm pore size filter. Then, NaCl and polyethylene glycol 8000 (PEG 8000) solution were added to reach respectively 1 M and 10% (wt/vol) final concentration to precipitate the phage particles. The solution was incubated overnight at 4 °C. The particles were collected by centrifugation of 27,720 g 30 min at 4 °C and suspended in 200 μL in LBP. Manufacturer's instructions were followed for the electrophoresis of phage particle samples using the Mini-PROTEAN TGXTM precast gels system (Bio-Rad, Hercules, CA, USA), followed by Coomassie blue staining to visualize precipitated proteins.

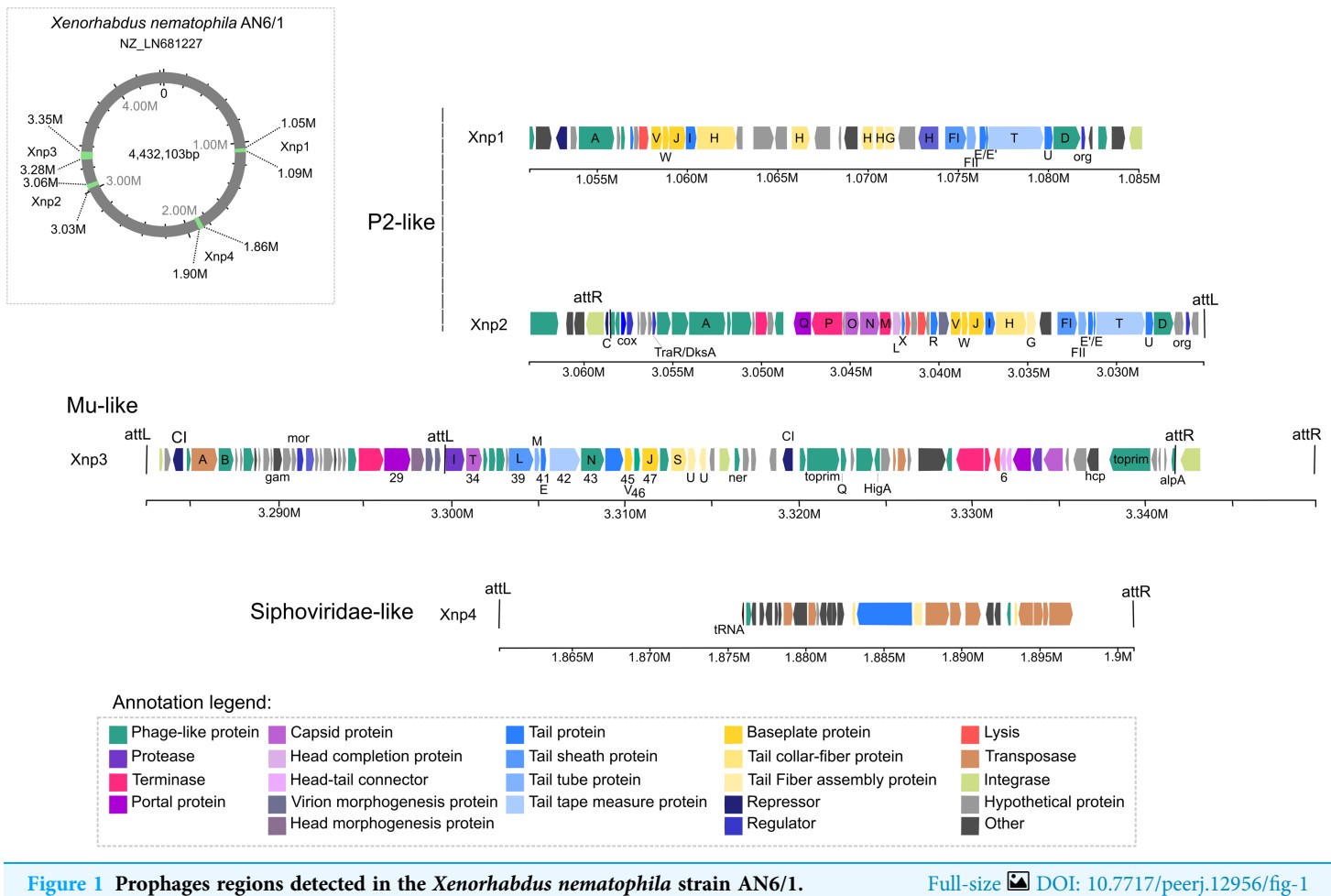

**Figure 1** Prophages regions detected in the *Xenorhabdus nematophila* strain AN6/1.

## RESULTS

### Four intact prophage regions are present in genomes of *X. nematophila* strain AN6/1

Of the 13 prophage regions identified by PHASTER in the studied genomes of *X. nematophila* strain AN6/1 (NZ_LN681227), only four were detected as potential "intact" prophage regions. Two of them are the prophage regions previously described: Xnp1 (localized at 1.05 M to 1.09 M), the P2-like remnant phage previously described as responsible of bacteriocin production; Xnp2 (localized at 3.03 M to 3.06 M), the P2-like complete phage previously described as not involved in bacteriocin production (Fig. 1). These two regions display similarities with P2-like bacteriophages. Our analysis also suggests strong similarities with the bacteriophage RE-2010 isolated from *Salmonella enterica Serovar Enteritidis* strain LK5 (NC_019488). As previously described, the prophage region xnp1 is characterized by the absence of genes coding head capsids proteins, as well as portal proteins or terminases. Another difference between the two P2-like prophages, is the region containing genes responsible to lysogeny control: Xnp2 region exhibited C immunity repressor and Cox protein while Xnp1 exhibited CI repressor (Fig. 1; Table S3).

The two newly identified prophage regions are "intact" regions and are designated as Xnp3 and Xnp4 to aligned with previously used nomenclature. Region Xnp3 (localized at 3.28 M to 3.35 M) presents strong similarities with Mu-like phage, while region Xnp4 (localized at 1.86 M to 1.90 M) is a remnant phage presenting similarities with Siphoviridae-like bacteriophages (Fig. 1). Compared to the other prophage regions, the Xnp4 region contained higher number of transposases and insertion sequence elements (IS), few phage-related proteins were clearly identified (only tail proteins) and no regulator or repressor protein was detected (Fig. 1; Table S3).

Our analysis also suggests strong similarities between Xnp3 and the SfMu bacteriophage isolated from *Shigella flexneri* (NC_027382) and the D108 bacteriophage isolated from *Escherichia coli* (NC_013594). PHASTER also detected another prophage region of 67.4 kb with two distinct prophage regions: (1) a 33.4 kb region (localized at 3.283 M to 3.316 M) which is like a complete Mu-like bacteriophage and (2) a 22.6 kb region (localized at 3.319 M to 3.341 M) which seems to be a remnant prophage (Fig. 1; Table S3). We focused on the first region (like Mu-like phage), which when compared to closely related bacteriophages, lacks the terminal part including one tail collar-fiber protein (S), as well as the DNA-binding transcriptional regulator Com and mom proteins (Fig. 2). This prophage region is not only present in *X. nematophila* but seems to be present (or has similar prophage regions) in the three other *Xenorhabdus* species including: *Xenorhabdus cabanillasii* strain JM26, *Xenorhabdus miraniensis* strain DSM17902 and *Xenorhabdus thuongxuanensis* strain 30TX1 (Fig. 2; Table S4).

## Characterization of effect of mitomycin C and UV exposure

Our results showed that bacterial grow is not disrupted by mitomycin C induction at early stages of induction (15 and 45 min) (Fig. 3). Contrastingly, both mitomycin C and UV triggered growth difference in both uninduced cultures and induced cultures in late induction stages (Fig. 3). Similarly, after 2 h, bacterial cells density was observed (a slightly lower with cultures exposed to UV), but after 10 h a high mortality rate was observed for the cultures exposed to mitomycin whereas a strong growth reduction was observed for the cultures exposed to UV (Fig. 3).

SDS-page electrophoresis showed the presence of phage-particles after 10 h in lysates of both mitomycin C induced cultures and UV-induced cultures (Fig. 4). We also observed four subunits more abundant: (1) below 37 kDa, (2) above 37 kDa, (3) above 50 kDa and (4) below 75 kDa. We speculate that the subunit above 37 kDa might be the outer membrane porin (OpnP) proteins based on previous reported data (*Kim et al., 2003*; *Leisman, Waukau & Forst, 1995*; *Morales-Soto & Forst, 2011*). Three other subunits (between 75–125 Dk) were detected but less abundant. We also observed low signal of phage-particles in lysate of mitomycin-induced cultures after 15 and 40 min of induction.

## Genetic mechanisms induced by mitomycin C and UV exposure

Regarding the level of gene expression after mitomycin C exposure, our preliminary analyses showed increase of the expression of *recA* and *lexA* at 45 min and 2 h after induction. We observed a down-regulation of *lexA* 10 h after both mitomycin C and UV
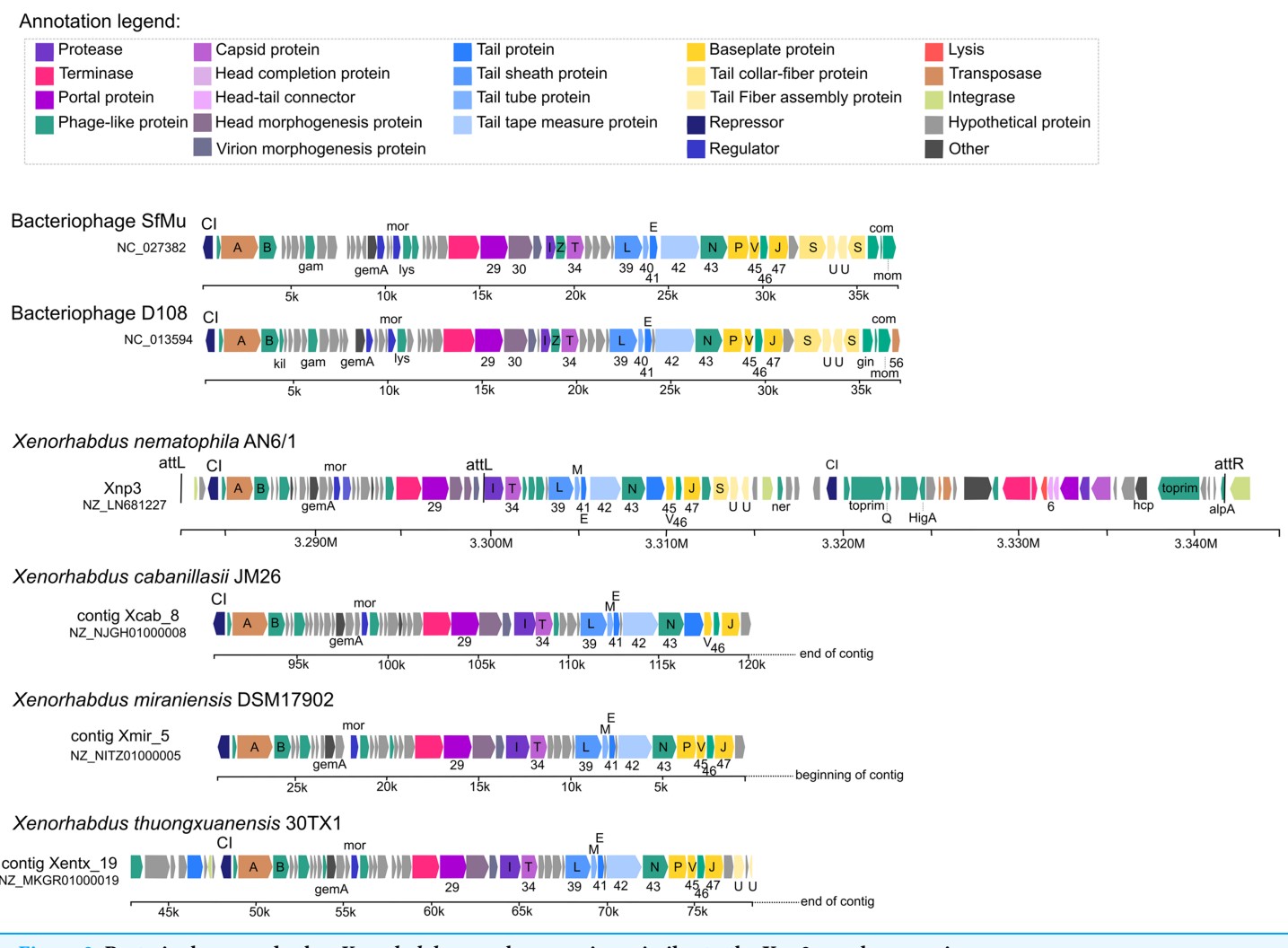

**Figure 2 Bacteriophages and other *Xenorhabdus* prophages regions similar to the Xnp3 prophages region.**

induction, while a down-regulation of *recA* 10 h after induction was only observed with UV induction (Fig. 5).

We were able to study the expression of the three prophages regions, Xnp1, Xnp3 and Xnp4 but the low level of expression of the Xnp2 did not allow presentation of robust data. It is consistent with previous studies, genes encoding the tube and sheath protein in Xnp2 prophage region were expressed at barely detectable levels (*Morales-Soto & Forst, 2011*).

We successfully quantified the relative expression of two genes in Xnp1 prophage regions, the replication gene, *GpA*, and the baseplate assembly protein W, *GpW*, as well as the putative CI repressor gene (Table S5). We observed up-regulation of *GpA* and *GpW* at 2 and 10 h after mitomycin exposure. With respect to UV exposure, the up regulation of these genes was observed only at 2 h post exposure. No significant differences were observed with the controls (uninduced cultures) at early time points. Regarding the CI repressor contained in the Xnp1 region, no significant differences were observed, however

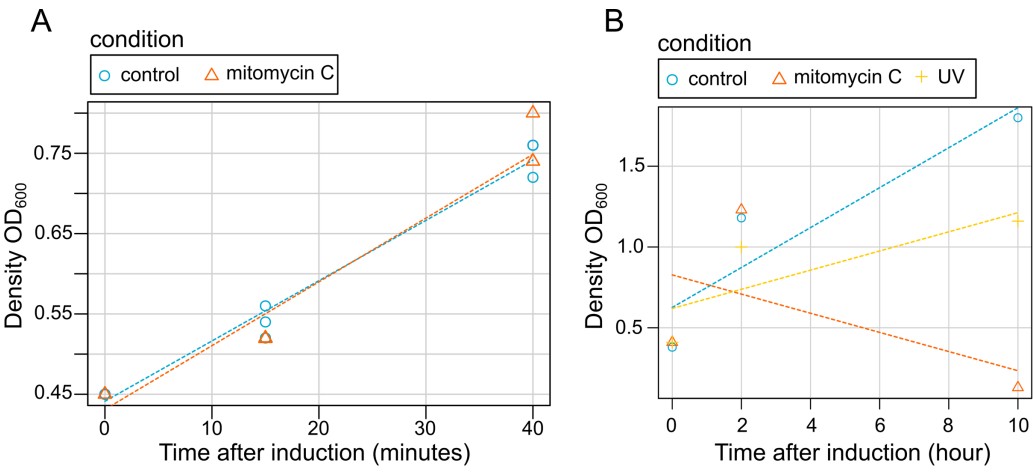

**Figure 3** **Effect of induction by mitomycin and UV exposure on growth of *X. nematophila* AN6/1 cultures.** (A) Optical density ($OD_{600}$) of *X. nematophila* AN6/1 cultures 15 and 40 min after mitomycin induction. (B) Optical density ($OD_{600}$) of *Xenorhabdus nematophila* AN6/1 cultures 2 and 10 h after mitomycin induction and UV induction. A fitted regression line for each condition was calculated by scatterplot (method = lm) in R environment.

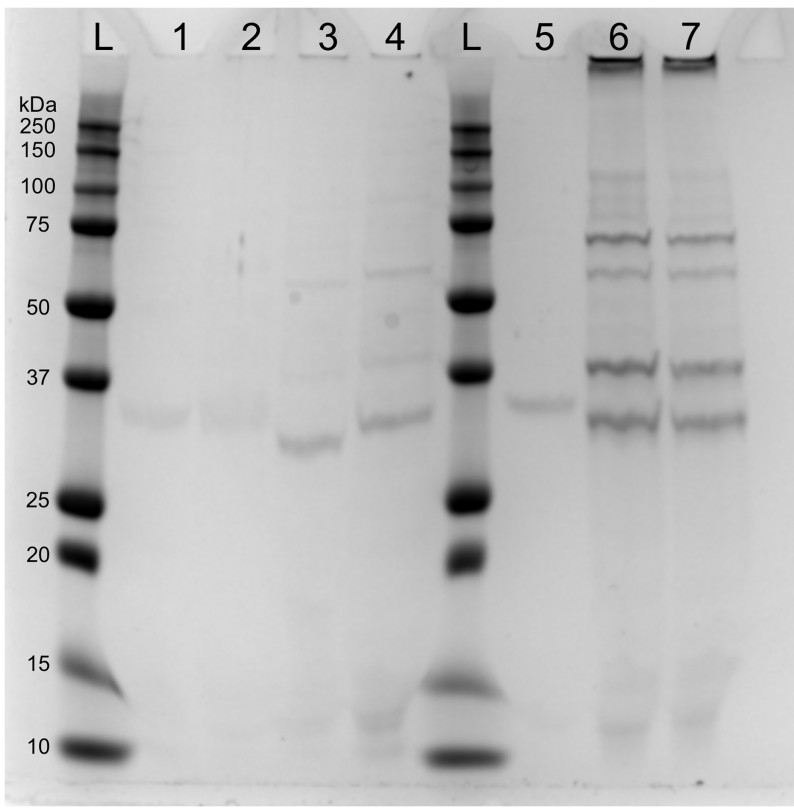

**Figure 4** **SDS-PAGE of lysate supernatant from *Xenorhabdus nematophila* AN6/1 cultures after PEG treatment.**

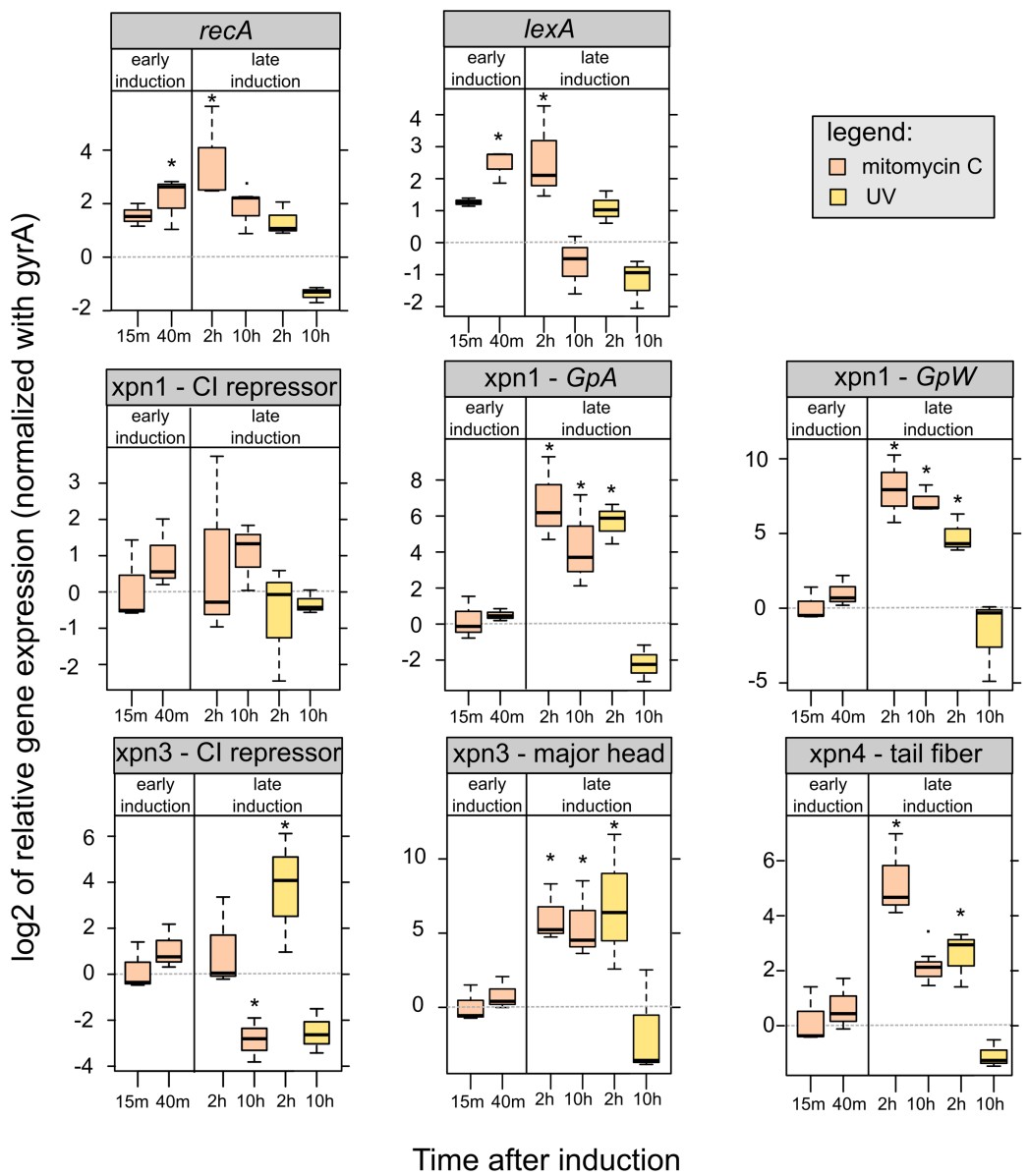

**Figure 5 Relative expression of *lexA*, *recA* and prophages genes of mitomycin C induced and UV-induced *Xenorhabdus nematophila* AN6/1 cultures.** Relative expression of each gene were normalized using gyrA as housekeeping gene and uninduced *Xenorhabdus nematophila* AN6/1 cultures were used as calibrator. The log base 2 of relative expression were plot and statistical tests were performed on relative expression value (see detail Table S5). Abbreviation: GpA, replication gene A; GpW, baseplate assembly protein W; CI, putative CI repressor gene. An asterisk (*) indicated that the statistical test showed significant difference with control. 

a tendency for up-regulation was observed in mitomycin-induced cultures (at 2 and 10 h post exposure) (Fig. 5).

Regarding the Xnp3 prophage regions, we quantified the relative expression of the major head protein, as well as the putative CI repressor (Table S5). We observed an up-regulation of the gene encoding the major head protein at 2 and 10 h post mitomycin C exposure; and an up-regulation only after 2 h for the UV exposure (Fig. 5). With respect to

the CI repressor, an up-regulation was observed after 2 h of UV exposure and thereafter, a down-regulation after 10 h was denoted for both mitomycin and UV exposure. However, no significant differential expression of the CI repressor was observed at early time post exposure (15 or 40 min).

With respect to Xnp4 prophage regions, we quantified the relative expression of the tail fiber protein (Table S5). We observed an up-regulation of the gene that encodes the tail fiber protein only 2 h post exposure to either mitomycin or UV. A down-regulation of the gene was observed at 10 h post-exposure for the UV-induced cultures only.

## DISCUSSION

In this study we expanded existing knowledge on the diversity of integrated phages in of *X. nematophila* genome by describing two additional prophage regions: Xnp3 and Xnp4. The previously described Xnp1 prophage region appears more conserved among *Xenorhabdus* spp. when compared to the other prophage regions (Fig. S1). The three other prophage regions were either not detected in the all the analyzed genomes *Xenorhabdus* species or presenting more variation. Our study showed that the newly described Xnp3 prophage region is observed in four genomes of *Xenorhabdus* species and is very similar to complete Mu-like bacteriophage. It is suggested that the acquisition of this prophage region is conserved among some *Xenorhabdus* species. Our study shows different levels of conservation of these prophage regions among *Xenorhabdus* species suggesting different selection pressures.

Regarding the mechanisms underlying the induction of phage-like particles in *Xenorhabdus*, it has been theorized that induced DNA damage could drive the cleavage regulation of the CI repressor present in the P2-remnant prophage locus (Xnp1) leading to the release of phage tail-like particles (*Morales-Soto et al., 2012*; *Thappeta et al., 2020*). Although DNA damage mechanisms have not been demonstrated for *Xenorhabdus* species, the network of genes known as SOS response appears as a near-universal mechanism in bacteria (*Fornelos, Browning & Butala, 2016*; *Lemire, Figueroa-Bossi & Bossi, 2011*; *Nanda, Thormann & Frunzke, 2015*). This gene regulatory network encompasses genes that are coordinately regulated by the repressor *LexA* (*Fornelos, Browning & Butala, 2016*). During DNA damage event, the gene *recA* is activate and stimulate the self-cleavage of LexA inducing DNA repair responses (*Fornelos, Browning & Butala, 2016*). In our study, we showed that *X. nematophila* cultures exposed to mitomycin C and UV exhibit an increase expression of *recA*, as well as *LexA*. These mechanisms appear to be activated early after exposure to mitomycin C, a slight increase was observed within 15 min postinduction. Interestingly, the UV induction condition used in our study (10 min exposure at 6 $\mu$J/cm$^2$) yielded a better recovery of the bacteria cultures than the mitomycin induction condition (concentration of 5 $\mu$g ml$^{-1}$) with lower mortality after 10 h of exposure despite the observed production of phage-particles. In this induction condition, the lower bacterial cell mortality was associated with the downregulation of the *recA* regulator (after 10 h). In both induction conditions, our study confirmed the involvement of *recA* and *lexA* in the DNA damage mechanisms driven by the induction.

Regarding the expression of the CI repressor, our results are less clear. For most of the conditions and time points no significant differential expression of the CI repressor for both Xpn1 and Xnp3 regions was observed. Although our results suggest low modification of the expression of the CI repressor after induction, further evidence is needed to provide a stronger support for these observations.

In this study we did not see evidence for the induction of Xnp2 prophage region by the host SOS response as previously reported (*Morales-Soto & Forst, 2011*). The Xnp2 genes were hardly detectable by qRT-PCR. It is known that not all integrated phages respond to the LexA system of induction as it has been demonstrated for Phage P2 integrated in *E. coli* (*Bertani & Bertani, 1970*). It has been demonstrated that the presence of gene coding the cox protein in the prophage region can negatively autoregulate the early operon inhibiting the formation of the lysogenic repressor (*Saha, Lundqvist & Haggård-Ljungquist, 1987*). The Xnp2 region contained a gene coding the cox regulator protein which could explained that this prophage region is not induced by mitomycin or UV exposure.

Our results showed that the genes contained in Xnp1 prophage region (*GpA* and *GpW*) are up regulated after mitomycin C and UV exposure. Similarly, *Morales-Soto & Forst (2011)* previously demonstrated an increase in the expression of genes contained in Xnp1 prophage region (*Morales-Soto & Forst, 2011*). In addition, our study showed that the Xnp1 prophage region is not the only prophage region up regulated during mitomycin C and UV exposure The expression of Xnp3 gene (encoding major head protein) and Xnp4 gene (encoding tail fiber protein) exhibited a very similar expression to those quantified for Xnp1 genes. Thus, an up-regulation of the three prophage regions was observed in *X. nematophila* during mitomycin C and UV exposures.

Previously, it has been shown that deletion of genes in Xnp1 region (coding sheath and fiber protein) in *X. nematophila* affect production of phage tail-like bacteriocin during mitomycin C induction (*Morales-Soto & Forst, 2011*). However, the phage tail-like bacteriocin or xenorhabdicin were not the only phage particles observed as produced by *X. nematophila* cultures exposed to mitomycin C, diverse phages particles were previously described in mitomycin C-induced cultures of *X. nematophila* (A24/1) (such as bacteriocin with extended sheath, empty sheath, empty phage head, or complete phages) (*Boemare et al., 1992*). According to our observations, the Xnp3 prophage region sequence, exhibited strong similarities with the complete Mu-like bacteriophage genome sequence. These similarities associated with results of up-regulation of genes contain in this prophage region after induction suggest that Xnp3 may be involved in phage-like particles, potentially complete phages. Similarly, we might speculate that the Xnp4 may be involved in other phage like-tail particles because only genes encoding tail and fiber protein were identified. However, further experiments including silencing of Xpn3 and Xnp4 genes is needed to corroborate this assumption.

## CONCLUSIONS

In this study two integrated prophage regions not previously described in the genome of *X. nematophila* AN6/1 were identified. Although less conserved among *Xenorhabdus* spp.

than the described p1 prophage region, we observed similar prophage region in other *Xenorhabdus* species. The Xnp3 prophage region very similar to complete Mu-like bacteriophage which was observed in three other available draft *Xenorhabdus* genomes. In addition, we outline potential genetic mechanisms that occur in *X. nematophila* AN6/1 during mitomycin C and UV exposure. We showed that mitomycin C exposure induced an up-regulation of *recA* and *lexA* suggesting activation of SOS response as described in other bacteria. Our findings suggest that mitomycin C and UV exposure lead to up-regulation of three on the four integrated prophages region present in *X. nematophila* genomes. Although our study did not allow for the recognition of an activation role of the newly described prophage region Xnp3 or Xnp4, it suggests that other prophage regions besides Xnp1 are inducible. Characterization of the induced particles using plaque formation experiment, as well as, TEM (Transmission Electron Microscopy) studies should be conducted to further characterized the described prophages. Further investigations are also warranted to better understand the diversity and biological role of integrate prophage regions in *Xenorhabdus species*.

## ACKNOWLEDGEMENTS

We acknowledge Dr. David Baltrus from University of Arizona and Dr. Lise Raleigh from New England Biolabs for their helpful comments on phage biology.

### Funding
This work was supported with start-up funds provided by the College of Agriculture and Life Sciences, University of Arizona to SPS. The funders had no role in study design, data collection and analysis, decision to publish, or preparation of the manuscript.

### Grant Disclosures
The following grant information was disclosed by the authors:
College of Agriculture and Life Sciences.
University of Arizona to SPS.

### Competing Interests
The authors declare that they have no competing interests.

### Author Contributions
- Emilie Lefoulon conceived and designed the experiments, performed the experiments, analyzed the data, prepared figures and/or tables, authored or reviewed drafts of the paper, and approved the final draft.
- Natalie Campbell performed the experiments, prepared figures and/or tables, and approved the final draft.
- S. Patricia Stock conceived and designed the experiments, analyzed the data, authored or reviewed drafts of the paper, and approved the final draft.

## Data Availability

The raw data are available in the Supplemental Files.

## Supplemental Information

Supplemental information for this article can be found online at http://dx.doi.org/10.7717/peerj.12956#supplemental-information.

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
