# Peer review of "Identification of novel prophage regions in Xenorhabdus nematophila genome and gene expression analysis during phage-like particle induction"

_PeerJ, doi:10.7717/peerj.12956_

## Round 0.1 · original submission · Minor Revisions

The manuscript was assessed by two experts in this area and found merit enough in your study, to be published in this journal, Please attend to the Reviewers' comments as suggested.

·

Basic reporting

Lefoulon and colleagues present an interesting study regarding the diversity of prophage regions in the Xenorhabdus nematophila genome and investigate the effect of SOS activating environmental cues that induce the expression of these prophages. The manuscript addresses an important aspect of many bacteria: their presence and impact on cell physiology and the environmental advantages of prophages.
Overall, the manuscript is well written and provides enough information regarding the prophages contained in this important organism. The finding is also relevant to the use of nematode-based plague control

Experimental design

Most of the experimental approaches used in this manuscript are sound and well designed. This reviewer finds one key experiment missing (see next section) and one important modification to Figure 4. Besides this, the manuscript is relevant, and the experimental design is well conducted and the data analysis.

Validity of the findings

This reviewer's only criticism regarding the manuscript is the lack of a very basic and simple experiment, plaque formation. This report contains important data, and I find it relevant. Still, any reader will question why authors are not doing phage dilutions and overlay platting so the reader can see that phage are present. Perhaps differences in plaque morphology, if the phages are forming clear or turbid plaques, may also suggest that the prophages identified may form plaques of different characteristics. This reviewer truly values the manuscript, but this is a key experiment to eliminate the readers' skepticism. Also, in Figure 4, do authors estimate the possible identity of the proteins based on molecular weight? I suggest adding at least the best estimate for a couple of proteins, indicating with an arrow, and proposing this in the figure legend.

Additional comments

This reviewer kindly requests that authors change the title of the manuscript; in my opinion, the work is extensive and complete, I strongly believe that the plaque formation assay is needed, but overall, the manuscript is a good characterization of the prophages found. I think the title may be 'Identification of novel prophage regions in Xenorhabdus nematophila genome and gene expression analysis during phage-like particle induction' by no means do I think this manuscript is preliminary. Perhaps the TEM analysis would be great to identify the phages and other experiments, but I think the data presented here is enough.
In the following lines, I provide some suggestions to the manuscript.
This is not mandatory, but in line 48-49, this reviewer somewhat disagrees with the statement included here, and many other authors have done regarding the energetic cost of prophages. Overall, the energy quote is somewhat neglectable since the genome does not overgrow with phages or remnants of them by insertion. In the end, the only negative effect is lysogenic particles that can become active. I kindly request the authors to reconsider this line.
Line 75, please remove an extra space in (respectively xnp1 and xbp1)
Line 99, space missing in 10 mL
Line 113, space missing un 750 µL
Line 116, missing letter, 10 min.
Line 117, space missing in 1 mL
Line 119, please correct to RNase-free water.
Line 143, space missing in 8 mL
Line 189, please remove the 'after' in this line.
In figure 3, the trend is the same in other independent experiments?
In line 270, please correct to E. coli
In line 273, please correct cox

Reviewer 2 ·

Basic reporting

The research was carried out with scientific rigor.
The structuring of the manuscript is adequate and all the raw data, so the supplementary material has access.

Experimental design

The methods used to achieve the objectives set out in the research are well described; however, there are some details that I will comment on shortly.

Validity of the findings

The suggested conclusions are supported by the results presented in the manuscript; however, I suggest you be more specific and concrete in this section, as I consider it to be too long.

Additional comments

The following corrections and suggestions are intended to improve the manuscript.

Line 38. mitomycin C
Line 40. ibidem.
Line 54. Xenorhabdus spp?.. complete with spp.
Line 55. Steinernema spp... ibidem.

As a suggestion, the content of lines 44-53 could be rearranged after line 61. This is in order to begin by introducing the reader to Xenorhabdus and its relationship to the host, and then place the text that refers to phages.

Line 72. Xenorhabdus spp.?
line 75. Complete the full name of X. bovienii
Line 79. mitomycin C
Line 79. X. bovienii in italic fount.
Line 80. ...gene of the Xnp1 or Xbp1 region... or is not in italic fount
Line 81-83. This part is a bit confusing, could you rewrite it?


Nothing is mentioned in the entire introduction about the rationale for choosing mitomycin C or UV light; What background is there on this?


Line 90. Xenorhabdus, what species?; nematophila?
Line 98. Xenorhabdus nematophila --> X. namatophia. In the rest of the text, the full name is no longer necessary.
Line 106. I suggest putting some amount in the agitation in rpm.
Line 108. mitomicyn C ?
Line 112. 5000 rpm I suggest changing to x g.
Line 113. Ibidem.
Line 116. 10 min?
Line 117. 13 000 rpm I suggest changing to x g.
Line. 118. Ibidem.
Lines 144-148. ibidem
Line 188. Complete mitomycin C
Line 189. Ibidem. And in the figure 3, the legend mitomycin C, also is important complete the name in the plots. And fig 5.
Lines 197-198. Dk --> kDa. Are you referring to kilodaltons? kDa is the correct

Change it, also in the gel image. Figure 4.

Line 210. In figure 5. legends with the GpA, Gpw and CI genes.

Line 233. Xenorhabdus spp?
Line. 302. X. nematophila
Line. 307. mitomicyn or mitomycin C?. Complete the full name in the text.

---

## Round 0.2 · Minor Revisions

The manuscript was modified following the Reviewers' suggestion and as a consequence, this version is significantly improved. The only point the authors were reluctant to address is the change of RPM units to x g. The latter is the internationally accepted unit to report relative centrifugal force and I agree with the Reviewer's comments and insist on modifying this parameter, for reproducibility purposes.

---

## Round 0.3 · accepted · Accept

I thank the authors for modifying the manuscript, and this version is now suitable for publication.